# Impact of maternal common mental disorders on child educational outcomes at 7 and 9 years: a population-based cohort study in Ethiopia

Habtamu Mekonnen,[1] Girmay Medhin,[2] Mark Tomlinson,[3] Atalay Alem,[1] Martin Prince,[4] Charlotte Hanlon[1,4]

[1]Department of Psychiatry, School of Medicine, College of Health Sciences, Addis Ababa University, Addis Ababa, Ethiopia
[2]Aklilu Lemma Institute of Pathobiology, Addis Ababa University, Addis Ababa, Ethiopia
[3]Department of Psychology, University of Stellenbosch, Stellenbosch, South Africa
[4]Health Services and Population Research Department, Institute of Psychiatry, Psychology and Neuroscience, King's College London, London, UK

**Correspondence to**
Dr Charlotte Hanlon;
charlotte.hanlon@kcl.ac.uk

## ABSTRACT

**Objectives** To examine the association between exposure to maternal common mental disorders (CMD) in preschool and early school age children and subsequent child educational outcomes.

**Design** A population-based cohort study.

**Setting** The study was undertaken in the Butajira health and demographic surveillance site (HDSS), a predominantly rural area of south central Ethiopia.

**Participants** Inclusion criteria are women aged between 15 and 49 years, able to speak Amharic, in the third trimester of pregnancy and resident of the HDSS. 1065 women were recruited between July 2005 and February 2006 and followed up. When the average age of children was 6.5 years old, the cohort was expanded to include an additional 1345 mothers and children who had been born in the 12 months preceding and following the recruitment of the original cohort, identified from the HDSS records. Data from a total of 2090 mother–child dyads were included in the current analysis.

**Measures** Maternal CMD was measured when the children were 6–7 (6/7) and 7–8 (7/8) years old using the Self-reporting Questionnaire, validated for the setting. Educational outcomes (dropout) of the children at aged 7/8 years (end of 2013/2014 academic year) were obtained from maternal report. At age 8/9 years (end of 2014/2015 academic year), educational outcomes (academic achievement, absenteeism and dropout) of the children were obtained from school records.

**Results** After adjusting for potential confounders, exposure to maternal CMD at 7/8 years was associated significantly with school dropout (OR 1.07; 95% CI 1.00 to 1.13, P=0.043) and absenteeism (incidence rate ratio 1.01; 95% CI 1.00 to 1.02 P=0.026) at the end of 2014/2015 academic year. There was no association between maternal CMD and child academic achievement.

**Conclusion** Future studies are needed to evaluate whether interventions to improve maternal mental health can reduce child school absenteeism and dropout.

## Strengths and limitations of this study

► To the best of our knowledge, this is the first study from a low-income country to investigate prospectively the association between maternal mental health and school absenteeism, dropout and academic performance in the child.
► Methodological strengths include the representative, population-based sample and large sample size.
► We relied on non-standardised tests of child academic achievement and on routinely collected data to measure absenteeism, which may have led to non-differential misclassification and underestimation of the strength of association.
► Diagnostic measures of common mental disorders would have been preferable to use of a screening scale.
► Chronic physical health problems in the child were not measured.

may not fully use their potential. An early nurturing environment is integral to establishing the solid foundations essential for child development, well-being and later academic success. Maternal common mental disorders (CMD), characterised by depressive, anxiety and somatic symptoms, may compromise early child development.[2] In a systematic review of studies from low-income and middle-income countries (LMICs), maternal CMD affected 15.6% of women during pregnancy and 19.8% during the postnatal period.[3] Evidence is accumulating from middle-income[4–6] and low-income[7–9] countries that the children of mothers with CMD have less optimal growth, cognitive and language development, even when taking social adversity into account. This association may be mediated through disruption of mother–child interactions[8 10] as has been seen in high-income country (HIC) settings.[11] In LMICs, maternal CMD has also been associated with low birth weight,[12] prolonged labour,[13] child ill-health[14]

## INTRODUCTION

Child education is a crucial component of holistic child development[1]; however, even when children have access to schooling, they

and child undernutrition,[15 16] all of which are established risk factors for poor child development.[17 18] Despite this emerging evidence base, to the best of our knowledge, there have been just two previous studies investigating the impact of maternal CMD on child educational outcomes, both of which were conducted in Barbados, which was at the time an upper middle-income country.[19 20] There have been no published studies from rural LMIC settings or low-income countries.

In this study, we examined the association between preschool and early school age exposure to maternal CMD and child educational outcomes in an ongoing population-based cohort study of mothers and children in rural Ethiopia. We hypothesised that children exposed to high symptoms of maternal CMD would have poorer school attendance, higher school dropout and poor academic achievement over 24 months of follow-up.

## METHODOLOGY
### Study design
The C-MaMiE study (child outcomes in relation to maternal mental health in Ethiopia) is a population-based cohort study.[13]

### Study setting
This study was undertaken in Butajira, Gurage Zone, Southern Nations Nationalities and Peoples' Region of Ethiopia. Butajira is predominantly a rural area, located 135 km south of the capital Addis Ababa, within a health and demographic surveillance site (HDSS). The HDSS was established in 1986 under the Butajira Rural Health Programme,[21] with one urban and nine rural administrative subdistricts from different ecological zones. In each subdistrict, there is an easily accessible health post (front-line primary care) and an elementary school. Mixed farming of cash crops, such as khat (*Catha edulis Forsk*) and chilli peppers, and production of staples, such as maize, 'false banana' or Ensete (*Ensete ventricosun*), form the basis of the economy.

### Context for education
In Ethiopia, primary education lasts for 8 years (age group 7–14 years) with two cycles: basic (grades 1–4) and general education (grades 5–8). In the first cycle, children are taught and evaluated by a single teacher following the 'self-contained class' concept.[22] Ethiopia is working towards complete coverage of primary education, yet only 85.5% has been achieved with 9.9% dropout and 7.3% grade repetition nationally.[23] The official age for school enrolment is 7 years.[22] All government schools are, in principle, free for all Ethiopians; however, families are expected to cover the costs of school uniforms, food and exercise books.[22] In the study area, private schools are only found in the urban (Butajira town) district, with fewer than 0.3% of study children enrolled in these schools. Except for two national examinations (at the completion of grades 10 and 12) and one regional examination (at the completion of grade 8), the academic performance of students is assessed by the class teacher using non-standardised tests.

### Study participants
The C-MaMiE birth cohort was established in the Butajira HDSS between July 2005 and February 2006 to investigate the public health impact of perinatal CMD.[13] A population-based sample of 1065 women was recruited out of 1234 eligible women (86.3%) with inclusion criteria of ages between 15 and 49 years, ability to speak Amharic, resident of the HDSS and in the third trimester of pregnancy. Fewer than 3% of women were excluded on the basis of language at baseline. The cohort of women and the child born from the index pregnancy have been assessed repeatedly over time. When the average age of the C-MaMiE project children was 6.5 years, the cohort size was augmented by 1345 children and their mothers (the 'expanded C-MaMiE cohort') who were identified by records in the HDSS as having been born in the 12 months preceding (n=572) and following (n=773) the recruitment of the C-MaMiE cohort. The same eligibility criteria were applied to the expanded cohort participants. In this study, 2090 mother–child dyads remain under follow-up and participated in the study.

### Assessment time points
Two assessments were conducted 12 months apart within the expanded C-MaMiE cohort; exposures were assessed first when the children were aged 6/7 years and repeated at age 7/8 years. Dropout of the children at age 7/8 years (end of 2013/2014 academic year) was obtained from maternal report, while at age 8/9 years (end of 2014/2015 academic year) the educational outcomes of absenteeism, dropout and academic achievement were obtained from school records.

### Measures
#### Outcomes
*School dropout* was operationalised as the proportion of students who had enrolled at the beginning of the academic year (September) but who had dropped out of school before the end of the academic year (June) and was obtained from school records and each mother. Children who drop out of school can be re-enrolled for the subsequent academic year.

*Academic achievement* was assessed using the child's grade point averaged over the two semesters of the Ethiopian school year. Grade repetition occurs when the averaged grade point is <50%.

*Absenteeism* was defined as the student missing school for a minimum of 1 day, irrespective of the reason given for the absence. In Ethiopia, each school keeps a daily attendance record for each student. For this study, the total number of days of absence from school was obtained from the school attendance sheet for the 2014/2015 academic year.

## Primary exposure

*Maternal CMD* was measured when the child was aged 6/7 and 7/8 years, using the WHO 20-item version of the Self-reporting Questionnaire (SRQ-20).[24] The SRQ-20 is a screening tool which asks about the existence or absence of depressive, anxiety and somatic symptoms in the preceding 1 month (answered 'yes' or 'no'). The SRQ-20 has been validated for perinatal women in this rural Ethiopian population.[25]

## Potential confounding factors

The following potential confounding factors were measured at both the 6/7-year and 7/8-year exposure time points:

*Stressful life events*: list of threatening experiences (LTE) measures 12 stressful life events over the preceding 6 months.[26] The LTE was translated into Amharic, adapted to local conditions and has been used in the C-MaMiE cohort study at all points of assessment.

*Socioeconomic status*: it was measured using self-report of current roof material (corrugated iron vs thatch), the experience of hunger in the preceding month due to lack of food or money and the existence of emergency resources for times of crisis.

*Paternal substance use:* maternal report of frequency of alcohol or khat use by the father.

*Demographic characteristics*: age of the mother, marital status, literacy level, birth order and sex of the child were obtained from self-report of the mother when the children were 6/7. The updated marital status of the woman was also recorded when the children were 7/8 years. The HDSS records were used to calculate the age of the child.

*Child nutritional status*: anthropometric measures were carried out by trained project data collectors. Weight was measured with digital floor scales. A stadiometer with a movable head piece was used for height. Using the WHO reference population,[27] weight-for-age z scores (underweight) and height-for-age z scores (stunting) were calculated using WHO Anthro software.[28] However, there was collinearity between height and weight and, therefore, height-for-age was included in the final model as it has been argued that it is a better summary measure of cumulative undernutrition.[29]

## Data management
### Data collection procedure

Interviews with the women and anthropometric measures of the child were carried out in the woman's home, or surrounding area, according to the woman's preference, to ensure privacy and confidentiality. Anthropometric measures were also conducted in school, when convenient. The project data collectors have all completed diploma level education and have been employed by the C-MaMiE project for the last 11 years. They are experienced in conducting interviews and in the use of the study measures. All data collectors received an additional 3 days of intensive training on the use of the instruments. Data collectors were not aware of the objectives of the study

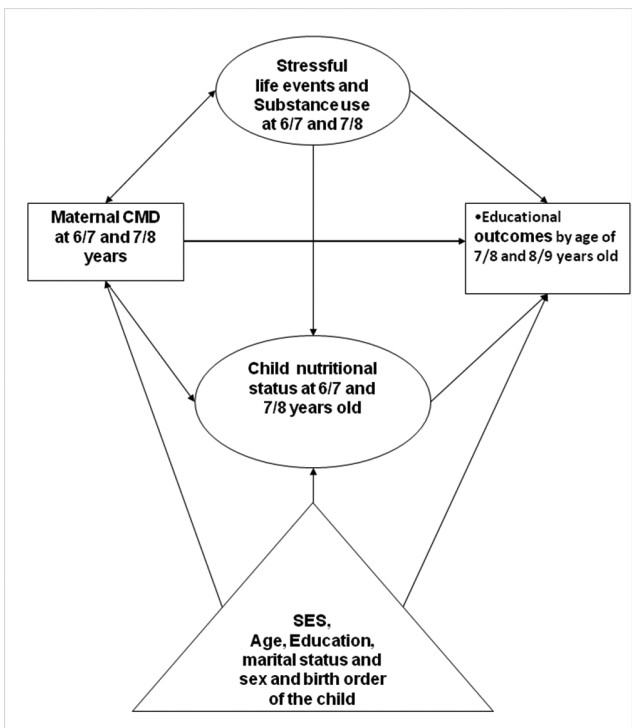

**Figure 1** Conceptual framework. CMD, common mental disorder; SES, socioeconomic status.

and interpretation of the score on the measures. Furthermore, they were randomly selected to carry out follow-up visits to specific households and time points were separated by at least 6 months, so the chance of bias due to recall of maternal CMD status was very low. The questionnaires were piloted before commencing data collection, and discrepancies in ratings were discussed to ensure that the data collectors had a common understanding.

### Maintaining data quality

Supervisors and a field coordinator monitored the data collection process and performed quality checks on a random sample of evaluations. Data were double entered with EpiData[30] by experienced data entry clerks on the day of data collection, where possible. Any identifiable information about the respondent was kept securely and separately detached from the assessment data and a code number was used to ensure confidentiality.

## Statistical analyses

The analysis was conducted using Stata software V.12.[31] The data were summarised using mean, median and percentage. A hypothesis-driven analysis was conducted to examine the association between maternal CMD (total score on the SRQ-20) and educational outcomes, guided by the conceptual model shown in figure 1. Initially, unadjusted analyses were conducted: logistic regression for school dropout and repetition of the year (binary), Poisson regression for absenteeism (continuous, count data) and linear regression for academic achievement (continuous, normally distributed). Multivariable analyses were then conducted, adjusted for all potential confounders identified a priori. Estimates of

association were presented with their corresponding 95% CIs. Complete case analysis was used. The study has been reported according to the Strengthening The Reporting of Observational Studies in Epidemiology (STROBE)-reporting checklist.

## ETHICAL CONSIDERATIONS

Consent from each woman and assent from the child were obtained. Any woman who presented with high symptoms of mental health problems and suicidal ideation was advised to seek care at the psychiatric unit at Butajira Hospital, with the project covering treatment and transportation costs.

## RESULTS

A total of 2090 mother–child dyads were assessed at 6/7 years, and 1957 were assessed at 7/8 years (see figure 2). Those who were lost to follow-up did not differ in terms of age, marital status, level of literacy, negative life event, socioeconomic status (SES), substance use or on their mental health status. There was no significant difference between those children who did and who did not have educational information available regarding maternal CMD, SES, parental literacy, substance use, negative life event, nutritional status, gender and their birth order.

See tables 1 and 2 for the distribution of maternal and child characteristics across educational outcomes.

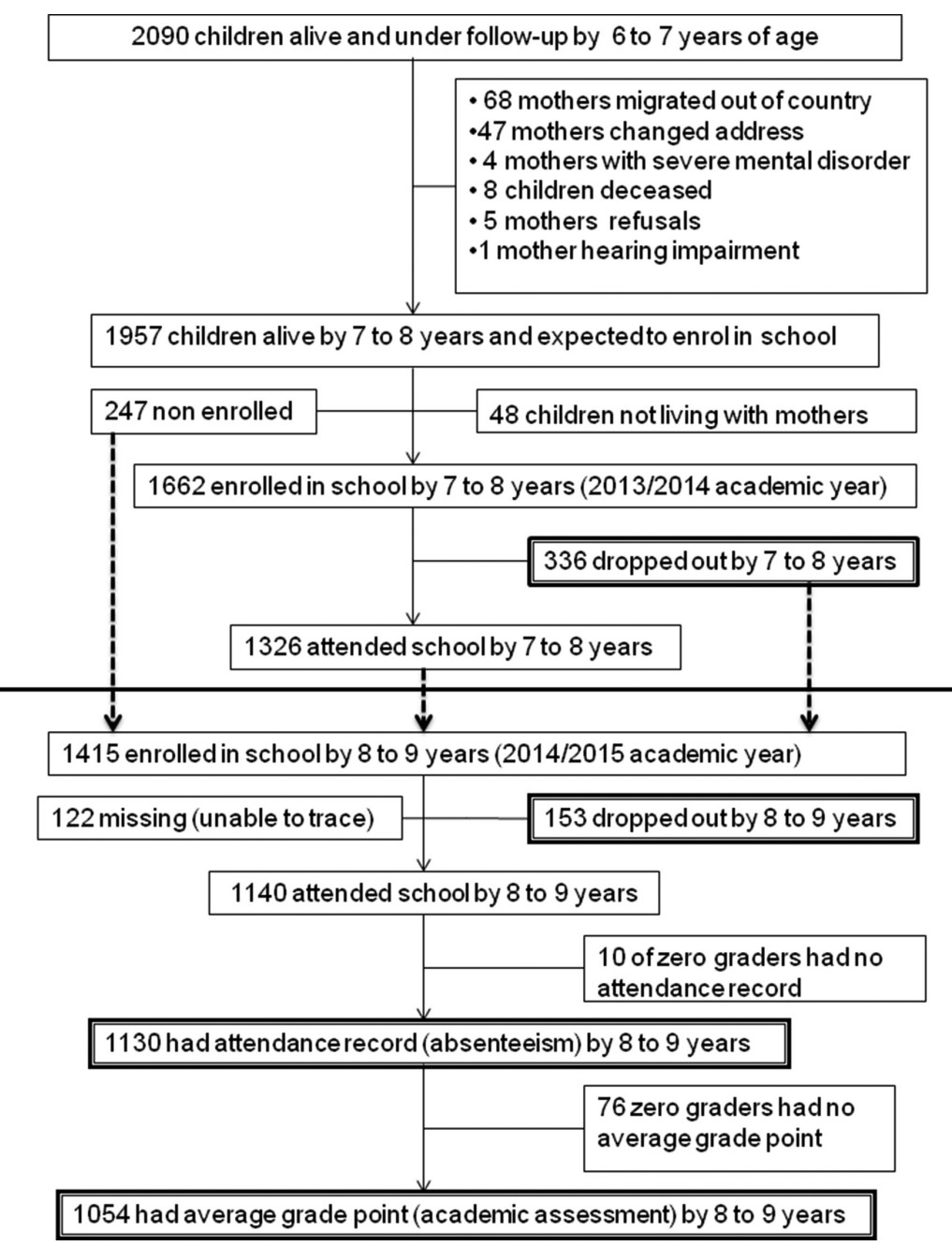

**Figure 2**  Follow-up chart for child educational outcomes.

**Table 1** Characteristics of participants in relation to child school dropout

| Exposure variables | | Exposure variables at the age of 6/7 years and child school dropout at the end of 2013/2014 academic year | | Exposure variables at the age of 7/8 years and child school dropout at the end of 2014/2015 academic year | |
|---|---|---|---|---|---|
| | | **Dropped out** | **Attending school** | **Dropped out** | **Attending school** |
| | | **n (%)** | **n (%)** | **n (%)** | **n (%)** |
| **Exposure variables** | | **336 (20.2)** | **1326 (79.8)** | **153 (11.8)** | **1140 (88.2)** |
| Parental characteristics | | | | | |
| Maternal age in years (mean, SD) | | 35 (6.00) | 34 (6.00) | 35 (6.00) | 34 (6.00) |
| Literacy (mother) | Non-literate | 307 (91.6) | 1100 (83.3) | 142 (93.4) | 997 (88.0) |
| Literacy (father) | Non-literate | 148 (46.7) | 423 (33.3) | 62 (42.5) | 400 (36.8) |
| Maternal marital status | Monogamous | 263 (78.5) | 1089 (82.4) | 116 (76.8) | 908 (81.0) |
| | Polygamous | 54 (16.1) | 182 (13.8) | 28 (18.5) | 160 (14.3) |
| | Divorced, widowed, separated | 18 (5.4) | 50 (3.8) | 7 (4.6) | 53 (4.7) |
| Socioeconomic status | | | | | |
| Current hunger | Had hunger | 42 (12.5) | 80 (6.1) | 12 (7.8) | 50 (4.4) |
| Emergency resources | No emergency resources | 164 (48.9) | 485 (36.7) | 49 (32.2) | 322 (28.4) |
| Roof material | Thatched roof | 242 (72.2) | 810 (61.3) | 94 (61.8) | 601 (53.0) |
| Psychosocial characteristics | | | | | |
| Paternal substance use | Weekly khat use | 264 (83.5) | 976 (76.9) | 104 (72.2) | 880 (82.7) |
| | Weekly alcohol use | 55 (17.4) | 185 (14.6) | 30 (20.8) | 173 (16.2) |
| Negative life event in the last 6 months | No life event | 272 (81.9) | 1065 (81.2) | 108 (71.1) | 860 (75.9) |
| | One life event | 48 (14.5) | 184 (14.0) | 31 (20.4) | 196 (17.3) |
| | Two or more | 12 (3.6) | 63 (4.8) | 13 (8.6) | 77 (6.8) |
| Child characteristics | | | | | |
| Childbirth order | First | 26 (7.7) | 212 (16.0) | 16 (10.5) | 156 (13.7) |
| | Middle or last | 310 (92.3) | 1114 (84.0) | 136 (89.5) | 977 (86.2) |
| Sex of the child | Boy | 201 (59.8) | 670 (50.6) | 85 (55.4) | 590 (52.0) |
| Child nutritional status | Stunted (height-for-age z score<−2) | 111 (33.4) | 297 (22.6) | 46 (30.3) | 261 (23.2) |

School dropout was 20.2% (n=336) at the end of the 2013/2014 academic year. At the end of the 2014/2015 academic year, dropout was 11.8% (n=153), the median number of days of absence was 5 (IQR 2–11), and the mean average grade point was 62.5 out of 100 (SD 9.21). The correlation of SRQ-20 score with absenteeism (Spearman's rank correlation coefficient=0.008; P=0.777) and averaged grade point (r=−0.039; P=0.208) was not significant. In the 2013/2014 academic year, the median score of SRQ-20 was 1, (IQR 0–2) in children who dropped out versus 0, (IQR 0–1) in those who stayed in school (P=0.003; Mann-Whitney test). Similarly, in 2014/2015 academic year, for inschool children the median SRQ-20 score was 0, (IQR 0–2) versus 1, (IQR 0–3.5) in those dropped out children (P=0.02; Mann-Whitney test).

Controlling for potential confounders, maternal CMD at 7/8 years was associated significantly with school dropout (OR 1.07, 95% CI 1.00 to 1.13) by the end of the 2014/2015 academic year, but the significant unadjusted association between maternal CMD at 6/7 years and school dropout by the end of the 2013/2014 academic year became non-significant (adjusted OR 1.05, 95% CI 0.99 to 1.12). In the fully adjusted multivariable model, parent non-literacy, low SES, male gender, not being first in the birth order and lower height-for-age were associated significantly with school dropout by the end of the 2013/2014 academic year. Paternal khat use and lower child height-for-age were associated significantly with 2014/2015 academic year school dropout. Maternal CMD at 7/8 years was associated with 2014/2015 academic

**Table 2** Characteristics of participants and child absenteeism, academic achievement and repetition of the school year

| | | Child school absenteeism at the end of 2014/2015 academic year | Child academic achievement at the end of 2014/2015 academic year | Child grade repetition status at the end of 2014/2015 academic year | |
| --- | --- | --- | --- | --- | --- |
| | | Median (25th, 75th percentiles) | Mean (SD) | Class repeated | Promoted to next grade |
| | | | | n (%) | n (%) |
| **Exposure variables at the age of 7/8 years** | | **5 (2, 11)** | **62.5 (9.21)** | **25 (2.4)** | **1029 (97.6)** |
| Parental characteristics | | | | | |
| Maternal age in years, mean (SD) | | | – | 34 (6.00) | 34 (6.00) |
| Literacy (mother) | Non-literate | 2 (5, 12) | 62.5 (9.29) | 22 (88.0) | 900 (87.9) |
| Literacy (father) | Non-literate | 3 (6, 14) | 61.3 (9.23) | 13 (54.2) | 351 (35.8) |
| Maternal marital status | Monogamous | 2 (5, 12) | 62.8 (9.34) | 19 (76.0) | 824 (81.4) |
| | Polygamous | 2 (4, 9) | 61.1 (8.91) | 5 (20.0) | 140 (13.8) |
| | Divorced, widowed, separated | 1 (4, 6) | 62.1 (8.13) | 1 (4.0) | 48 (4.7) |
| Socioeconomic status | | | | | |
| Current hunger | Had hunger | 2 (5, 11) | 62.1 (7.75) | 1 (4.0) | 44 (4.3) |
| Emergency resource | Had no emergency resource | 2 (5, 11) | 62.1 (9.07) | 9 (36.0) | 288 (28.1) |
| Roof materials | Thatched | 3 (6, 14) | 62.8 (9.65) | 17 (68.0) | 524 (51.2) |
| Psychosocial characteristics | | | | | |
| Paternal substance use | Weekly khat use | 2 (5, 12) | 62.3 (9.24) | 16 (66.7) | 791 (82.4) |
| | Weekly alcohol use | 1 (3.5, 7) | 64.6 (9.42) | 6 (25.0) | 150 (15.6) |
| Negative life event in the last 6 months | No life event | 2 (5, 12) | 62.5 (9.14) | 20 (80.0) | 775 (75.6) |
| | One life event | 2 (4, 9) | 62.8 (9.84) | 3 (12.5) | 180 (17.6) |
| | Two or more | 2 (3, 8) | 61.2 (8.21) | 2 (8.0) | 68 (6.6) |
| Child characteristics | | | | | |
| Childbirth order | First | 2 (4.5, 14) | 62.8 (8.87) | 5 (20.0) | 136 (13.3) |
| | Middle or last | 2 (5, 11) | 62.5 (9.26) | 20 (80.0) | 887 (86.7) |
| Sex of the child | Boy | 2 (5, 12) | 62.6 (9.23) | 15 (60.0) | 528 (51.6) |
| Child nutritional status | Stunted (height-for-age z score<−2) | 2 (5, 12) | 62.5 (9.20) | 6 (24.0) | 233 (23.0) |

year school absenteeism (adjusted incidence rate ratio 1.01, 95% CI 1.00 to 1.02). Absenteeism was also associated significantly with maternal age, parent non-literacy, widowed or separated marital status, low SES, mother experiencing negative life event, child male gender and being first in the birth order. See tables 3 and 4.

There was no significant association between maternal CMD and academic achievement or grade repetition. Paternal non-literacy, alcohol use and widowed or separated maternal marital status were significantly associated independently with these outcomes. See table 5.

## DISCUSSION

We found a significant prospective association between maternal CMD and child school dropout and absenteeism. The unadjusted association between exposure to maternal CMD at age 6/7 years and dropout by the end of the 2013/2014 academic year became non-significant when adjusted for potential confounding variables. Parental low SES, non-literacy and substance use were associated significantly with adverse educational outcomes of children. There was no association between maternal CMD and child academic achievement or grade repetition.

As far as we are aware, our study is the first of its kind from a low-income country to report the association between exposure to preschool maternal CMD and subsequent adverse educational outcomes. Strengths of the study included the population-based design, large sample size, high follow-up rates and use of culturally validated measures. Nonetheless, there were limitations of our study. Maternal CMD was assessed using a symptom screening scale rather than a diagnostic measure. We assessed academic achievement

**Table 3** Factors associated with child school dropout

| Exposure variables | Exposure variables at the age of 6/7 years and school dropout at the end of 2013/2014 academic year OR (95% CI) | | Exposure variables at the age of 7/8 years and school dropout at the end of 2014/2015 academic years OR (95% CI) | |
|---|---|---|---|---|
| | Crude | Adjusted* | Crude | Adjusted* |
| **Primary exposure** | | | | |
| Maternal CMD (SRQ-20 total score) | **1.07 (1.02 to 1.13)** | 1.05 (0.99 to 1.12) | **1.07 (1.01 to 1.13)** | **1.07 (1.00 to 1.13)** |
| **Other explanatory variables** | | | | |
| Maternal age (years) | **1.03 (1.01 to 1.05)** | 1.02 (0.99 to 1.03) | 1.01 (0.98 to 1.04) | 1.01 (0.97 to 1.05) |
| Non-literate mother | **2.20 (1.46 to 3.33)** | **1.77 (1.13 to 2.77)** | 1.94 (0.99 to 3.76) | 1.64 (0.82 to 3.27) |
| Non-literate father | **1.75 (1.37 to 2.25)** | **1.53 (1.17 to 2.01)** | 1.26 (0.89 to 1.78) | 1.19 (0.82 to 1.74) |
| Maternal marital status (polygamous, divorced and widowed) | 1.08 (0.99 to 1.18) | 1.07 (0.82 to 1.43) | 1.03 (0.91 to 1.18) | 1.18 (0.78 to 1.78) |
| Experienced hunger due to lack of resources | **2.22 (1.50 to 3.29)** | **1.74 (1.10 to 2.77)** | 1.85 (0.96 to 3.56) | 1.78 (0.82 to 3.84) |
| No emergency resources | **1.65 (1.30 to 2.11)** | **1.43 (1.10 to 1.89)** | 1.20 (0.83 to 1.72) | 1.05 (0.69 to 1.60) |
| Thatched roof | **1.65 (1.27 to 2.14)** | 1.30 (0.96 to 1.74) | **1.43 (1.01 to 2.03)** | 1.33 (0.91 to 1.96) |
| Paternal khat use at least weekly | **1.52 (1.10 to 2.11)** | 1.38 (0.98 to 1.95) | **0.54 (0.37 to 0.81)** | **0.52 (0.34 to 0.79)** |
| Paternal alcohol use at least weekly | 1.23 (0.89 to 1.71) | 1.35 (0.95 to 1.93) | 1.35 (0.88 to 2.09) | 1.18 (0.73 to 1.89) |
| Negative life event (≥1 in the last 6 months) | 0.99 (0.84 to 1.16) | 0.85 (0.69 to 1.03) | 1.04 (0.98 to 1.89) | 1.00 (0.77 to 1.29) |
| Male child | **1.45 (1.14 to 1.85)** | **1.50 (1.15 to 1.96)** | 1.15 (0.82 to 1.62) | 1.13 (0.78 to 1.63) |
| Child first in birth order | **0.44 (0.29 to 0.67)** | **0.59 (0.36 to 0.95)** | 0.73 (0.43 to 1.27) | 0.82 (0.45 to 1.51) |
| Height-for-age z score | **0.73 (0.65 to 0.83)** | **0.80 (0.70 to 0.91)** | **0.75 (0.62 to 0.89)** | **0.73 (0.60 to 0.89)** |

*Parental characteristics include maternal age, marital status, maternal and paternal level of literacy, socioeconomic status (hunger due to lack of resources, emergency resources, roof material), paternal substance use, negative life event, child sex, birth order and child nutritional status.
Bold font is used to indicate that the association is statistically significant at a level of p<0.05.
CMD, common mental disorder; SRQ, Self-reporting Questionnaire.

using a composite and non-standardised measure (individual teacher assessment). Although this approach increased measurement error, composite measures which include participation, homework, attendance and tests given by teachers may be more ecologically valid and tied to day-to-day routine of teaching and learning than narrowly focused assessments of content mastery. Absenteeism was extracted from data collected routinely by schools, which is likely to have led to underestimation of absences, but this would have led to a non-differential error which would be expected to reduce the chance of finding a significant association and would not be expected to have led to bias. We relied on proxy indicators of SES. Although these indicators have been developed for the population under study, they relied on self-report from the women and may not have been sufficiently comprehensive, thus raising the possibility of residual confounding. Physical ill-health of the child may also have confounded the association.

The higher school dropout rate at the end of 2013/2014 academic year over the subsequent year is in keeping with existing evidence. The non-significant association between maternal CMD and academic achievement and grade repetition in our study is in contrast to the study from Barbados, where postpartum depression was associated with poorer academic achievement at 11 years of age.[20] This difference may have arisen due to the measures of academic achievement used (standardised in Barbados vs subjective/composite measure in Ethiopia) and the timing of the CMD exposure (postpartum vs preschool and early school years). Furthermore, the broader context for learning also differs substantially between Barbados and Ethiopia. In Ethiopia, the majority of women are non-literate and there is limited availability of learning resources (eg, books, games) within the home. Given the critical importance of these factors for child learning, any additional effect of maternal CMD may have been too small to detect, unlike in Barbados.

On the other hand, the deficits in the home learning environment mean that regular attendance at school

**Table 4** Factors associated with child school absenteeism

| Exposure variables at the age of 7/8 years of the child | School absenteeism at the end of 2014/2015 academic years Incidence rate ratio (95% CI) | |
| --- | --- | --- |
| | Crude | Adjusted* |
| Primary exposure | | |
| Maternal CMD (SRQ-20 total score) | 1.00 (0.99 to 1.01) | **1.01 (1.00 to 1.02)** |
| Other explanatory variables | | |
| Increasing maternal age (years) | 1.00 (0.99 to 1.01) | **1.01 (1.01 to 1.02)** |
| Mother non-literate | **1.60 (1.48 to 1.74)** | **1.57 (1.44 to 1.72)** |
| Father non-literate | **1.18 (1.13 to 1.23)** | **1.07 (1.02 to 1.12)** |
| Marital status (polygamous, divorced, widowed) | **0.92 (0.89 to 0.94)** | **0.88 (0.83 to 0.93)** |
| Experienced hunger due to lack of resources | 1.06 (0.96 to 1.17) | 1.06 (0.94 to 1.19) |
| No emergency resources | 1.03 (0.99 to 1.08) | 0.98 (0.94 to 1.04) |
| Thatched roof (vs corrugated iron) | **1.46 (1.40 to 1.53)** | **1.36 (1.30 to 1.42)** |
| Father uses khat at least weekly | **1.10 (1.03 to 1.17)** | 1.04 (0.97 to 1.11) |
| Father uses alcohol at least weekly | **0.90 (0.85 to 0.96)** | 0.92 (0.86 to 0.98) |
| Negative life event (≥1 in the last 6 months) | **0.98 (0.97 to 0.99)** | **0.90 (0.87 to 0.94)** |
| Male child | 1.04 (1.00 to 1.09) | **1.05 (1.00 to 1.09)** |
| Child first in birth order | **1.14 (1.08 to 1.21)** | **1.17 (1.10 to 1.25)** |
| Height-for-age z score | 1.00 (0.97 to 1.01) | 1.00 (0.97 to 1.02) |

*Parental characteristics include (maternal age, marital status, maternal and paternal level of literacy), socioeconomic status (hunger due to lack of resources, emergency resources, roof material), paternal substance use, negative life event, child sex, birth order and child nutritional status.
Bold font is used to indicate that the association is statistically significant at a level of p<0.05.
CMD, common mental disorder; SRQ, Self-reporting Questionnaire.

becomes the most important way for a child to learn and is likely to have greater influence on child educational performance. Our findings were in keeping with the Barbados study, where depressive symptoms in mothers of a child with an early history of malnutrition were associated with school absences between 5 and 11 years of age. Irregular school attendance is likely to be on the pathway to poorer educational attainment and to identify a group of children who are at risk of fully dropping out from school.

In studies from HICs, erratic school attendance and dropout from school have been associated with an adverse early home environment, living with smokers, the quality of early care-giving, SES, cognitive development, behaviour problems, academic achievement, peer relations and parent involvement.[32 33] In LMICs, food insecurity and death of parents[34 35] are associated with irregular school attendance and dropping out entirely. In both HIC and LMIC studies, maternal CMD could be an important unmeasured mediator of the effects of adverse home environment on school attendance. For children under the age of 10 years, absenteeism and dropout are unlikely to happen without the knowledge of the family, and in particular the mother.

School absenteeism and dropout have important economic implications, particularly in a low-income country like Ethiopia. Repeating years of education or students taking longer to complete their education

brings inefficiency into the school system, financial burden on the household and may adversely affect the motivation of the child to pursue their education. For those students who drop out of school altogether, there are economic losses from negative impacts on future productivity. Every additional year of schooling has a substantial positive effect on adult wages.[36] Achieving high coverage of primary school enrolment is undermined by subsequent early dropout from schooling, which will limit Ethiopia's ability to deliver on the Sustainable Development Goal target to ensure that all girls and boys complete free, equitable and quality primary and secondary education by 2030.

School-based interventions targeting specific predictors of absenteeism and dropout in LMICs, for example, deworming,[37] hand wash campaigns, safe water and hygiene,[38] and school feeding programmes,[39] have been found to be effective in reducing absenteeism and dropout. Our study indicates that interventions (at family, school or community level) focusing on the reduction of absenteeism and dropout need also to incorporate maternal CMD. There is a global impetus to improve access to primary care-based mental healthcare in LMICs through the WHO's mental health Gap Action Programme.[40] Evidence-based treatment packages for maternal depression that can be delivered by general health workers (assessment and prescription of medication) and lay workers (psychosocial approaches) have been shown to be effective, feasible and

**Table 5** Factors associated with child academic achievement

| Exposure variables at the age of 7/8 years of the child | Academic achievement by the end of 2014/2015 academic year β coefficient (95% CI) | | Grade repetition by the end of 2014/2015 academic year OR (95% CI) | |
|---|---|---|---|---|
| | Crude | Adjusted* | Crude | Adjusted* |
| Primary exposure | | | | |
| Maternal CMD (SRQ-20 total score) | –0.07 (–0.27 to 0.13) | 0.01 (–0.22 to 0.23) | 1.01 (0.87 to 1.16) | 0.99 (0.85 to 1.16) |
| Other explanatory variables | | | | |
| Maternal age | –0.03 (–0.13 to 0.08) | 0.01 (–0.11 to 0.13) | 1.01 (0.94 to 1.09) | 1.05 (0.96 to 1.14) |
| Mother non-literate | 0.21 (–1.51 to 1.92) | 0.46 (–1.42 to 2.34) | 1.00 (0.29 to 0.97) | 0.87 (0.24 to 3.17) |
| Father non-literate | **–1.88 (–3.06 to 0.69)** | **–1.90 (–3.16 to –0.64)** | 2.12 (0.93 to 4.78) | 1.89 (0.80 to 4.44) |
| Marital status (polygamous, divorced and widowed) | –0.21 (–0.67 to 0.26) | **–2.02 (–3.48 to –0.55)** | 0.96 (0.67 to 1.38) | 1.31 (0.54 to 3.16) |
| Experienced hunger due to lack of resources | –0.42 (–3.18 to 2.33) | –0.79 (–4.14 to 2.56) | 0.93 (0.12 to 7.00) | 0.97 (0.11 to 8.74) |
| No emergency resources | –0.59 (–1.83 to 0.64) | –0.59 (–2.00 to 0.80) | 1.43 (0.63 to 3.28) | 1. 50 (0.60 to 3.74) |
| Thatched roof | 0.54 (–0.57 to 1.66) | 0.78 (–0.46 to 2.00) | 2.02 (0.86 to 4.73) | 1.55 (0.62 to 3.89) |
| Father uses khat at least weekly | –1.31 (–2.82 to 0.21) | –0.57 (–2.15 to 1.00) | 0.43 (0.18 to 1.01) | 0.44 (0.18 to 1.12) |
| Father uses alcohol at least weekly | **2.47 (0.88 to 4.05)** | **2.35 (0.65 to 4.04)** | 1.80 (0.70 to 4.62) | 1. 26 (0.45 to 3.51) |
| Negative life event in the last 6 months | –0.07 (–0.33 to 0.18) | –0.24 (–1.09 to 0.61) | 1.04 (0.90 to 1.18) | 0.76 (0.34 to 1.72) |
| Male child | 0.16 (–0.95 to 1.28) | –0.09 (–1.10 to 1.27) | 1.41 (0.63 to 3.16) | 1.45 (0.61 to 3.42) |
| Child first in birth order | 0.34 (–1.30 to 1.97) | 0.33 (–1.50 to 2.17) | 1.63 (0.60 to 4.41) | 2.09 (0.66 to 6.60) |
| Height-for-age z score | **0.62 (0.00 to 1.23)** | 0.59 (–0.07 to 1.25) | 1.08 (0.69 to 1.67) | 1.12 (0.69 to 1.81) |

*Parental characteristics include maternal age, marital status, maternal and paternal level of literacy, socioeconomic status (hunger due to lack of resources, emergency resources, roof material), paternal substance use, negative life event, child sex, birth order and child nutritional status.
Bold font is used to indicate that the association is statistically significant at a level of p<0.05.
CMD, common mental disorder; SRQ, Self-reporting Questionnaire.

acceptable to communities across LMIC settings.[41] Effective care for maternal depression has been shown to have beneficial effects on child health in Pakistan,[42] but further studies are needed to evaluate the impact on child educational outcomes.

## CONCLUSION

In this population-based cohort study in rural Ethiopia, children exposed to mothers with symptoms of common mental disorders during preschool and early school age were at greater risk of school dropout and absenteeism. Future studies are needed to understand the mechanisms underlying this association. Current global efforts to expand access to mental healthcare offer an opportunity to address maternal CMD as a key component of programmes to increase school attendance.

**Acknowledgements** We are grateful to the women who participated in the study for giving their time and energy to complete interviews and to the staff in the schools who facilitated data collection process. We also thank the project data collectors and staff of the C-MaMiE project in Butajira.

**Contributors** HM, CH, AA and MP conceptualised the study. HM led data collection with oversight from CH, AA and GM. HM led data analysis, with input from CH, AA, MP, MT and GM. HM drafted the manuscript and all other coauthors revised it critically. All authors approved the final version of the paper and agree to be accountable for all aspects of the work.

**Funding** This study was funded by the Wellcome Trust (project grant: 093559).

**Competing interests** None declared.

**Patient consent** Obtained.

**Ethics approval** Ethical approval was obtained from the Institutional Review Board of the College of Health Sciences, Addis Ababa University, Ethiopia (reference number 082/13/psy) and the Research Ethics Committee of King's College London, UK (reference number PNM/13/14-92).

**Provenance and peer review** Not commissioned; externally peer reviewed.

**Data sharing statement** The data are being used for a PhD student (HM) for his thesis and are not, therefore, available at the present time to the general public. The data may be requested from the corresponding author for verification of the analyses in this paper.

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
