## [Reviewer comments · BMJ Open]

ARTICLE DETAILS

TITLE (PROVISIONAL)	Impact of maternal common mental disorders on child educational outcomes at 7 and 9 years: a population-based cohort study in Ethiopia
AUTHORS	Mekonnen, Habtamu; Medhin, Girmay; Tomlinson, Mark; Alem, Atalay; Prince, Martin; Hanlon, Charlotte

VERSION 1 – REVIEW

REVIEWER	Abiodun Adewuya Lagos State University College of Medicine. Lagos. Nigeria
REVIEW RETURNED	14-Aug-2017

GENERAL COMMENTS	This is a timely and important paper evaluating the long term impact of maternal common mental disorders on this child child. This time not on child growth and development but on educational outcomes. This cohort had been well followed and further boosted by an additional 13445 women and children born in the 12 months preceding and following the recruitment of the original cohort COMMENTS 1. It will be nice to know if child chronic illnesses (asthma, epilepsy, sickle cell etc) or acute illnesses (malaria, respiratory tract infection etc) were considered as either confounders or mediators or were these reflected in the List of Threatening Experiences (LTE)?2. Were all the children attending the "free" government schools?..were there some of the children attending private schools?.3. is it possible that using the project data collectors who have been employed from same environment, and have been collecting data from same cohort for 11 years might bias the assessment?...Would they have been able to remember which mother/child scored well/poorly on which scale at the last assessment?. Both the positive and negative sides of this should be mentioned CONCLUSION on the whole i must congratulate the team for this work and it will surely add to the body of knowledge
--

REVIEWER	Khaled Ismail University of Birmingham, UK
REVIEW RETURNED	17-Aug-2017

GENERAL COMMENTS	This manuscript address an important clinical question that has wider implications. The manuscript is well written in general. However, some sections of the abstract are vague with some typos and grammatical errors that obscure the meaning, e.g.: Page 2 line 33 " when the CHILD was 6.5" - do the authors mean the age of the first child born in their original cohort? or is this the mean age of the children... In fact the sentence in page 7 lines 35 and 36 is a slightly better description but still not optimal. Also page 2 line 48 is confusing. Please revise the abstract conclusion. The study demonstrated a significant association between CMD and school drop-out and absence from school for some age groups, however, the study did not demonstrate that "interventions to improve maternal mental health may have benefits for child education." Please review sentence in page 4 line 5 "This prospective study has shown that children who exposed to maternal CMD during pre" and page 11 lines 35-36 "Those where were lost to follow-up did not differ in terms of demographic...." The finding that the "univariate association between maternal CMD at 6/7 years and school drop-out by 7/8 years became non-significant (adjusted OR 1.05, 95% CI: 0.99, 1.12)." needs to be clarified in the discussion and conclusion. The authors raise an interesting point in the discussion about the possible explanation of the variation in prevalence of CMD at 6/7 and 7/8. This section wold be complemented by a discussion about the implications of the higher prevalence on such an analysis particularly with regards to the sample size. Is it possible that the difference was not significant at 7/8 because the sample size was not large enough to demonstrate a difference at a lower prevalence. It is also interesting that the frequency of drop out at 8/9 was much lower than 7/8.
---

REVIEWER	Ahad Mahmud Khan Johns Hopkins University - Projahnmo Bangladesh
REVIEW RETURNED	19-Aug-2017

GENERAL COMMENTS	This article sought to address an important issue in public mental health. The objectives were to examine the association between maternal CMD and child educational outcomes i.e., absenteeism, drop out and poor academic achievements. Revisions needed: Page 2. Abstract - Need clarification what time point maternal CMD was measured. Page 4 Introduction - As the term 'common mental disorders' is not a diagnostic term it is necessary to define it more elaborately with references.
--

	Page 7. Study participants - Please elaborate the sampling technique and how was the sample size determined? Page 8. Need more clarification of operational definition of academic achievement. 'Missing school for a minimum of ONE DAY in an academic year' might not be logical to operationalize absenteeism. Also need to mention the cut-off score to define CMD with references. Page 9. Socioeconomic status was measured using roof material, experience of hunger and emergency resources. Please add the reference if any in favor of it. It will be more logical if wealth index can be calculated or any standardized SES scale can be used. Page 19. SRQ-20 is a screening tool; not a diagnostic one. It can underestimate or overestimate the prevalence of CMD. It should be mentioned in limitations. Although you did adjust for some confounding factors, residual confounding is still likely (it is difficult to capture all relevant factors) and so this needs to be recognized as a limitation as well.
--	--

REVIEWER	Laura Schwab Reese Purdue University, USA
REVIEW RETURNED	01-Sep-2017

GENERAL COMMENTS	Thank you for the opportunity to review this manuscript, which examined the role of maternal common mental disorders on children's educational outcomes. Although much of my work focuses on maternal mental health, my outcome of interest is not related to education. As such, I am providing the review of the statistical methods, reporting, and interpretation for this manuscript. I have several concerns in the structure and reporting of the analysis, which I believe must be addressed before the manuscript is suitable for publication. My overarching comment is that the authors have not presented their findings in a way that is easily accessible to the reader. There were many areas that I struggled to follow. I have provided comments that I hope will help the authors address this concern. Comments related to the methodology section 1. Study participant section a. As many of your readers will not be familiar with the Butajira HDSS, it would be helpful to understand how the inclusion criteria influenced the sample selection. Is pregnancy common outside of the ages of 15 and 49? What proportion of the women in the community speak Amharic? b. Did the expanded C-MaMiE cohort follow the same inclusion criteria (e.g., age, ability to speak Amharic, resident of the HDSS at the time of the birth?) If not, it will be important to discuss how these differences may impact the cohort. c. Although 1,234 women participated in the first cohort and 1,345 women participated in the expanded cohort, it is not clear from this section how many participants were a part of the study for the time points included in this analysis. It would be helpful to explain the cohort as it relates to this specific study, rather than the current general description of the study.
---

2. Assessment time-points section

a. The current language used to describe the time points is confusing since the phrase “7 to 8 (7/8) years is used to describe both an exposure time point and an outcome time point, but I believe it references two different time points. My understanding is that the 7/8 year exposure time point refers to a mother/child assessment that occurred at some point when the child was 7 or 8, but the 7/8 outcome time point occurs at the end of the school year to assess past year educational outcomes. Further clarification/editing of the results may reduce this issue, but it is currently confusing in the result section. Alternate phrasing, such as “2014 Academic Year” or “1st End of School Assessment” for outcomes and “1st Maternal Report” or “2014 Maternal/Child Assessment” may provide further clarity to the results.

3. Potential mediators section

a. Please provide a citation to support these anthropometric measures as an indicator of child nutritional status. It is not clear how weight is included in the models, as only height is mentioned in the tables.

4. Statistical analysis section

a. Regression models are not univariate statistics. “Crude” or “unadjusted” would be a more appropriate way to describe the methods mentioned on page 10, lines 43-48.

b. Given the substantial number of covariates identified a priori, it may be beneficial to statistically assess the appropriateness of their inclusion using statistical methods, especially considering issues like collinearity. Bias-variance tradeoff is another approach to statistically assess the appropriateness of the covariates.

c. The method of exploring mediation is not robust. The work by David Kenny or Tyler VanderWeele outlines more robust ways of approaching mediation analysis. Any mention of the mediation should be removed or the analysis should be updated.

Comments related to the results

1. Table 1--It is not clear how the authors arrive at the sample size in table 1. It would be helpful to describe in a footnote how participants were excluded from this analysis.

2. Table 2—It would be helpful to use consistent language between the table and the text. For example, “Child School absenteeism by the age of 8/9 years” was referenced as “the number of days of absence” in the text. It would also be helpful to have an indication, in the tables, what the numbers represent (e.g., what does a 5 mean in absenteeism?).

3. There are some other issues with language consistency in tables 3 and 4.

REVIEWER	William Turner PhD School for Policy Studies University of Bristol UK
REVIEW RETURNED	14-Sep-2017

GENERAL COMMENTS	I really enjoyed reading your work; this is a well-conducted and reported study which, I believe, will make a significant contribution to an emerging body of literature on MH issues in LMICs. Introduction: the section provides succinct description of the rationale and need for this population-based cohort study of mothers and children in rural Ethiopia aiming to examine the association between exposure to maternal CMD and child educational outcomes. Methodology: I could not identify any issues with the methodological parameters of the study; the section provides sufficient details in relation to study design, setting of study, sample, sampling, confounders and mediators, outcome measures & assessment points, data management and data analytic approach. Results: characteristics of participants (Tables 1 & 2) and findings from inferential analyses (Tables 3 & 4) are clearly and concisely presented. Discussion: the section provides a measured and balanced discussion of the significant (and non-significant associations (e.g. CMD and children's academic achievement), with good reference to the economic implications of the findings for a low-income country like Ethiopia. Though I concur with the authors' recommendation that further research would be needed to understand the mechanisms underlying the link between CMDs and child educational outcomes, esp. in different geographical contexts, the study highlights the need to consider integration of mental health interventions into community programmes aiming to improve educational outcomes for children, esp. in LMICs. The study offers a significant contribution to an emerging body of literature and offers useful insights (and recommendations) for future research. I have no hesitation recommending its publication to the journal. Thank you for giving me the opportunity to review this important work.
---

VERSION 1 – AUTHOR RESPONSE

Reviewer 1:

Comments

1. It will be nice to know if child chronic illnesses (asthma, epilepsy, sickle cell etc) or acute illnesses (malaria, respiratory tract infection etc) were considered as either confounders or mediators or were these reflected in the List of Threatening Experiences (LTE)?

Response: Diagnosed child chronic illness is rare in this cohort (asthma 0.4%, wheeze 2.0%, epilepsy < 1%, HIV < 1% in adult population, sickle cell does not affect this population). Acute infectious illness is common in this population. We did not measure acute illness and have included this as a limitation. However, we expect that only recurrent infectious illness would be relevant for educational outcomes and that this would mostly be captured within the child's body mass index.

2. Were all the children attending the "free" government schools?..were there some of the children attending private schools?

Response: In the study area more than 99% schools are government owned, but in one of the district (the Butajira town) there are a small number of recently established private schools. Fewer than 0.3% of children are enrolled within these private schools. We have now clarified this in the paper.

3. Is it possible that using the project data collectors who have been employed from same environment, and have been collecting data from same cohort for 11 years might bias the assessment?...Would they have been able to remember which mother/child scored well/poorly on which scale at the last assessment?. Both the positive and negative sides of this should be mentioned.

Response: We have now mentioned both the potential positive and negative sides of data collector retention. The likelihood of data collectors recalling specific scores from the study participants is low because of the following reasons: (1) the data collectors were not aware of the objectives of the study and the interpretation of the scores on each scale, the focus were only on the quality of administration. (2) data collectors were assigned randomly to a household using a lottery method (for fairness because some of the homes are difficult to access) which reduced the chance that they would be visiting the same home on consecutive time-points, and (3) there was a minimum of six months gap between each time point assessment and the data collectors were administering a whole battery of measures, so they are unlikely to have remembered performance on specific scales. Rather, as noted, the retention of data collectors brought many benefits, including acceptability with the participants, knowing how to locate homes and familiarity with the questionnaires.

Reviewer 2:

Comments:

1. The manuscript is well written in general. However, some sections of the abstract are vague with some typos and grammatical errors that obscure the meaning, e.g.: Page 2 line 33 " when the CHILD was 6.5" - do the authors mean the age of the first child born in their original cohort? or is this the mean age of the children... In fact the sentence in page 7 lines 35 and 36 is a slightly better description but still not optimal.

Response: We meant that we expanded the cohort when the children of the original cohort were 6.5 years old on average. This has now been clarified.

2. Also page 2 line 48 is confusing.

Response: We have rewritten this sentence.

3. Please revise the abstract conclusion. The study demonstrated a significant association between CMD and school drop-out and absence from school for some age groups, however, the study did not demonstrate that "interventions to improve maternal mental health may have benefits for child education."

Response: We have rewritten the conclusion as follows:

Future studies are needed to evaluate whether interventions to improve maternal mental health can reduce child school absenteeism and drop-out.

4. Please review sentence in page 4 line 5 "This prospective study has shown that children who exposed to maternal CMD during pre"

Response: This section has now been removed from the manuscript upon the recommendation of the editor, as it is not the requirement of the journal.

5. ...and page 11 lines 35-36 "Those where were lost to follow-up did not differ in terms of demographic...."

Response: this has now been clarified.

6. The finding that the "univariate association between maternal CMD at 6/7 years and school drop-out by 7/8 years became non-significant (adjusted OR 1.05, 95% CI: 0.99, 1.12)." needs to be clarified in the discussion and conclusion.

Response: We have ensured that this is more clearly presented and discussed.

7. The authors raise an interesting point in the discussion about the possible explanation of the variation in prevalence of CMD at 6/7 and 7/8. This section would be complemented by a discussion about the implications of the higher prevalence on such an analysis particularly with regards to the sample size. Is it possible that the difference was not significant at 7/8 because the sample size was not large enough to demonstrate a difference at a lower prevalence. It is also interesting that the frequency of drop out at 8/9 was much lower than 7/8.

Response: If maternal CMD had been used as a binary primary exposure in the models, this might have an implication for the sample size, but we used it as a continuous exposure. We have now stated, however, the general limitations of using a screening scale over diagnostic in the revised version.

With regard to drop-out, the finding of lower drop-out at higher age groups is consistent with the literature. We have now made note of this in the revised manuscript.

Reviewer 3

1. Page 2. Abstract - Need clarification what time point maternal CMD was measured.

Response: We have sought to clarify this.

2. Page 4 Introduction - As the term 'common mental disorders' is not a diagnostic term it is necessary to define it more elaborately with references.

Response: We have clarified the definition and provided a reference to the classic work on common mental disorders.

3. Page 7. Study participants - Please elaborate the sampling technique and how was the sample size determined?

Response: The sample size was restricted by the size of the cohort. The original cohort was a population-based sample. We included all available cohort members in the current analyses. We have made amendments to Figure 2 to clarify the participants contributing to each analysis and added clarifications to the methods section.

4. Page 8. Need more clarification of operational definition of academic achievement. 'Missing school for a minimum of ONE DAY in an academic year' might not be logical to operationalize absenteeism. Also need to mention the cut-off score to define CMD with references.

Response: There are different approaches to operationalization of absenteeism in the literature, with some studies using a cut-off point (to identify chronic absence) while others use the total days of absence. We followed the second approach and have used absenteeism as a continuous dependent variable.

5. Page 9. Socioeconomic status was measured using roof material, experience of hunger and emergency resources. Please add the reference if any in favor of it. It will be more logical if wealth index can be calculated or any standardized SES scale can be used.

Response: There is no standardized measure of socio-economic status in Ethiopia and the combination of urban and rural populations in the study sample makes it challenging to use asset or wealth indices. The indices that we used have been used in the Butajira HDS; however, we acknowledge the potential inadequacy of the measures in the limitations.

6. Page 19. SRQ-20 is a screening tool; not a diagnostic one. It can underestimate or overestimate the prevalence of CMD. It should be mentioned in limitations. Although you did adjust for some confounding factors, residual confounding is still likely (it is difficult to capture all relevant factors) and so this needs to be recognized as a limitation as well.

Response: we have expanded consideration of these limitations and removed presentation of prevalence estimates of maternal CMD.

Reviewer 4:

1. As many of your readers will not be familiar with the Butajira HDSS, it would be helpful to understand how the inclusion criteria influenced the sample selection. Is pregnancy common outside of the ages of 15 and 49? What proportion of the women in the community speak Amharic?

Response: There is no formal data on the pregnancy outside of the range of 15 and 49 years in the area. Above the age of 49 years is likely to be extremely rare. Child marriage is a crime within the constitutional law of the country. Thus, we anticipate that the effect of age restriction would have been negligible.

Regarding speaking Amharic, as mentioned in the study setting, Amharic is the official language in the district, language of instruction in school and language of market place in the community, so almost all inhabitants of the Butajira HDS can communicate in Amharic. At baseline, fewer than 3% of potentially eligible women were excluded on the basis of language. We have added this clarification to the methods section.

2. Did the expanded C-MaMiE cohort follow the same inclusion criteria (e.g., age, ability to speak Amharic, resident of the HDSS at the time of the birth?) If not, it will be important to discuss the how these difference may impact the cohort.

Response: yes, the expanded cohort participants were included on the basis of the same criteria. That was the benefit of being able to use the HDS database.

3. Although 1,234 women participated in the first cohort and 1,345 women participants in the expanded cohort, it is not clear from this section how many participants were a part of the study for the time points included in this analysis. It would be helpful to explain the cohort as it relates to this specific study, rather than the current general description of the study.

Response: we have revised this section and amended Figure 2 (the flow chart of participants) to make it clearer.

4. The current language used to describe the time points is confusing since the phrase "7 to 8 (7/8) years is used to describe both an exposure time point and an outcome time point, but I believe it references two different time points. My understanding is that the 7/8 year exposure time point refers to a mother/child assessment that occurred at some point when the child was 7 or 8, but the 7/8 outcome time point occurs at the end of the school year to assess past year educational outcomes. Further clarification/editing of the results may reduce this issue, but it is currently confusing in the result section. Alternate phrasing, such as "2014 Academic Year" or "1st End of School Assessment" for outcomes and "1st Maternal Report" or "2014 Maternal/Child Assessment" may provide further clarity to the results.

Response: we appreciate your suggestions and have used them to try to make this clearer throughout the manuscript.

5. Please provide a citation to support these anthropometric measures as an indicator of child nutritional status. It is not clear how weight is included in the models, as only height is mentioned in the tables.

Response: We have provided a citation to support the anthropometric measures used. There was collinearity between weight and height; therefore, as height-for-age is considered to be a better indicator of chronic undernutrition and more relevant to educational outcomes, we used height-for-age in the model.

6. Regression models are not univariate statistics. “Crude” or “unadjusted” would be a more appropriate way to describe the methods mentioned on page 10, lines 43-48.

Response: We have amended as per the recommendations.

7. Given the substantial number of covariates identified a priori, it may be beneficial to statistically assess the appropriateness of their inclusion using statistical methods, especially considering issues like collinearity. Bias-variance tradeoff is another approach to statistically assess the appropriateness of the covariates.

Response: We were aware of this potential problem and checked variables for collinearity as suggested; hence, leading to exclusion of weight-for-age in the model.

8. The method of exploring mediation is not robust. The work by David Kenny or Tyler VanderWeele outlines more robust ways of approaching mediation analysis. Any mention of the mediation should be removed or the analysis should be updated.

Response: We accept the comment and have opted to drop the discussion of possible mediation as the association between maternal CMD and child educational outcomes was not affected by the presence or absence of child nutritional status in the model.

9. Table 1--It is not clear how the authors arrive at the sample size in table 1. It would be helpful to describe in a footnote how participants were excluded from this analysis.

Response: We have amended Figure 2 to clarify the sample sizes involved in each analysis.

10. Table 2—It would be helpful to use consistent language between the table and the text. For example, “Child School absenteeism by the age of 8/9 years” was referenced as “the number of days of absence” in the text. It would also be helpful to have an indication, in the tables, what the numbers represent (e.g., what does a 5 mean in absenteeism?).

Response: we have amended accordingly.

11. There are some other issues with language consistency in tables 3 and 4.

Response: these have now been resolved.

Reviewer 5:

1. Though I concur with the authors’ recommendation that further research would be needed to understand the mechanisms underlying the link between CMDs and child educational outcomes, esp. in different geographical contexts, the study highlights the need to consider integration of mental health interventions into community programmes aiming to improve educational outcomes for children, esp. in LMICs.

Response: Reviewer 2 had concerns about us over-stepping the limits of our data in our recommendations, so we have been deliberately cautious.

Response to comment from the JECH reviewer

1. The SRQ-20 is generally used as a screening questionnaire for depression. The authors mention that this questionnaire has been validated in a population of Ethiopian women in the perinatal period, but fail to point out that this study showed that the SRQ was not good at detecting 'cases'. The validation study (Hanlon et al. 2008) states " The utility of SRQ in detecting 'cases' of CMD was not established, with differing estimates of optimal cut-off score: three and above in Study 1 (sensitivity 85.7%, specificity 75.6%); seven and above in Study 2 (sensitivity 68.4%, specificity 62%)." In view of this, it seems inappropriate to use a cut-point of seven or above in the current study to derive estimates of prevalence of maternal CMD at two time-points, to base descriptive statistics on those prevalence estimates, and to discuss why prevalence might have changed over time. The validation paper suggested that the SRQ-20 had more use as a continuous measure of symptoms of CMD. In the current paper, the authors use the SRQ-20 as a continuous measure in linear regression analyses, but given that it is likely to be highly skewed in its distribution, with most women scoring 0 or 1 (not details are provided in the paper), I think this is inappropriate.

Response: as mentioned by the reviewer, the Ethiopia SRQ-20 validation study that we cited does indicate validity of the SRQ-20 as a continuous measure of common mental disorder symptoms. The SRQ-20 was only used as a continuous variable in the regression analyses because of these concerns with the cut-off. As the SRQ-20 was included in the model as an exposure variable, the non-normal distribution did not affect the regression model assumptions. We have removed the description of the prevalence of CMD from the results and the discussion. We have added discussion of the limitations of using a screening rather than diagnostic tool.

2. My other major concern with this paper is the use of a measure of academic achievement that is unstandardized and based on composite measures that is likely to vary between teachers.

Response: we have included this as a limitation.

3. Other minor points relate to a lack of information in the methods section. I would have liked to know how discrepancies between maternal and school reports of drop out were reconciled, how was literacy assessed, and what were the response options for the question about roofing material.

Response: The handling of discrepancies between maternal and school report of school drop-out is now discussed in the revised document. Maternal literacy was assessed by evaluating the woman's ability to read and understand the information form. Paternal literacy relied on self-report of the woman. The response options for roofing material were corrugated iron and thatch. We have added this information to the methods.

VERSION 2 – REVIEW

REVIEWER	Ahad Mahmud Khan Johns Hopkins University - Projahnmo Bangladesh
REVIEW RETURNED	06-Nov-2017

GENERAL COMMENTS	The responses are fine for me. I have no further comment.
---

REVIEWER	Laura Schwab Reese Purdue University, USA
REVIEW RETURNED	28-Nov-2017

GENERAL COMMENTS	The authors have sufficiently addressed my concerns.
--